# Do Not Go Gentle into That Deaf Night: A Holistic Perspective on Cochlear Implant Use as Part of Healthy Aging

**DOI:** 10.3390/jpm12101658

**Published:** 2022-10-05

**Authors:** Angelika Illg, Julia Lukaschyk, Eugen Kludt, Anke Lesinski-Schiedat, Mareike Billinger-Finke

**Affiliations:** 1German Hearing Center, Department of Otorhinolaryngology, Medical University Hannover, Karl-Wiechert-Allee 3, 30625 Hannover, Germany; 2MED-EL GmbH, 6020 Innsbruck, Austria

**Keywords:** cochlear implant, elderly, patient-reported outcomes measures, speech perception, cost-effectiveness, NCIQ

## Abstract

Research suggests that cochlear implant (CI) use in elderly people improves speech perception and health-related quality of life (HRQOL). CI provision could also prevent dementia and other comorbidities and support healthy aging. The aim of this study was (1) to prospectively investigate potential changes in HRQOL and speech perception and (2) to identify clinical action points to improve CI treatment. Participants (*n* = 45) were CI recipients aged 60–90 with postlingual deafness. They were divided into groups, according to age: Group 1 (*n* = 20) received a CI between the age of 60–70 years; group 2 (*n* = 25) between the age of 71–90 years. HRQOL and speech perception were assessed preoperatively, and three and twelve months postoperatively. HRQOL and speech perception increased significantly within one year postoperatively in both groups. No difference between groups was found. We conclude that CI treatment improves speech perception and HRQOL in elderly users. Improvement of the referral process for CI treatment and a holistic approach when discussing CI treatment in the elderly population could prevent auditory deprivation and the deterioration of cognitive abilities.

## 1. Introduction

The average life expectancy is currently estimated to be >80 years for a newborn baby in Germany [1]. Due to the demographic change in Germany, the number of people aged 70 years and older increased from 8 million in 1990 to 13 million in 2019 [2] and the amount of people of working age (51.8 million in 2018) is expected to decrease by 4–6 million by 2035 [3]. This change will increase health care expenditure [4] and likely force an increase in the age at which Germans may retire with full pension. Therefore, it is important that the aging population receive the health care they need to live longer (working) lives [5].

Epidemiological studies demonstrated that approximately 30% of men and 20% of women in Europe have at least 30 dB hearing loss (HL) by the age of 70 years; this increases to 55% of men and 45% of women by the age of 80 years [6]. Hearing loss currently affects around 20% of the global population, which is equivalent to 1.5 billion people [7]. Previous work has demonstrated that even a slight hearing loss of 10 dB makes it twice as difficult to communicate with other people, particularly at social events or at work [8].

The majority of adults with hearing-impairment have sensorineural hearing loss (SNHL), which is characterized by non-functional or absent hair cells. Cochlear implants (CI) are the gold standard intervention for most people with severe-to-profound or profound SNHL [9]. CIs can restore hearing by directly stimulating the auditory nerve electrically. Several studies and reviews have concluded that CI use improves the speech perception ability and health-related quality of life (HRQOL) of people with SNHL [9,10,11,12].

As of 2017, only 50,000 of the more than 1 million people in Germany who are considered CI candidates have received one [9]. According to Figure 1, people who receive a CI are equally distributed across age. However, hearing loss prevalence increases strongly with age [5,7], which is not surprising, as its most common cause is related to the process of aging [13]. Only one third (32%) of people who received a CI in Germany in 2019 were elderly (65 years or older) and only 15% were 75 years or older (Figure 1). Importantly, this indicates that CIs are underutilized, particularly within the elderly population. As summarized by D’Haese et al. [13], the World Health Organization (WHO) has estimated that worldwide only 10% of adults with severe to profound hearing loss have received a CI. Similarly, insufficient CI treatment in relation to CI candidates has been estimated from Sweden [14] and Japan [15].

When considering CI treatment in elderly people, several aspects need to be considered that may not need to be considered with children or younger adults. The following paragraphs will therefore highlight CI treatment and outcomes particularly in the elderly population.

Several studies have demonstrated that CI use has benefits beyond increased hearing abilities [16,17,18]. CI treatment in elderly people reduces tinnitus; depression; and somatization disorders; and, importantly in the context of elderly CI recipients, decreases loneliness, arrests cognitive decline and even increases cognitive functions, and increases HRQOL [19,20,21,22,23,24,25,26,27,28,29,30,31].

In a recent report, Knopke et al. [32] postulated that the outcomes in elderly CI users depend on their psychological status and demonstrated that, after 12 months of CI use, anxiety and depressive symptoms correlated negatively with HRQOL in users aged 70 to 88 years. Despite the presence of physical comorbidities (25% in group 1 and 40% in group 2), HRQOL increased and reached the same level in both groups after twelve months of CI use [32].

It is important to consider how age, cognition, and auditory rehabilitation impact each other. Some studies have reported that elderly CI users have shallower learning curves in speech perception than adult users under 65 years of age and therefore have suggested elderly users, who benefit from receiving age-targeted rehabilitation [33,34,35]. Importantly, untreated hearing loss is one big reversible factor for dementia [7,36]. A recent publication concluded that CIs are safe and effective in people with mild cognitive impairment [37]. Approximately 60% of the subjects screened had mild cognitive impairment, which had decreased slightly but significantly at the 6-month CI aftercare appointment. These findings are in line with the conclusion that CI treatment could facilitate the concept of healthy aging in elderly people [38,39].

People aged 65 years are more likely than other age groups to have comorbidities, often have multimorbidities, and may have additional handicaps [40]. Published data suggest that CI treatment in the elderly is safe and does not lead to an increase in complications [5,41,42]. Still, when considering the elderly and people with more and more complex comorbidities for CI treatment, it is also important to consider the risks associated with general anesthesia, which increase with age and comorbidities. This has led to discussions regarding performing CI surgery under local anesthesia. A recent systematic review and meta-analysis concluded that CI treatment under local anesthesia is safe [36].

Lastly, studies have investigated the cost-effectiveness of CI treatment with its political, economic, and ethical aspects. A quality-adjusted life year (QALY) can quantify the benefits of CI treatment as a function of HRQOL and time duration, hence it is directly linked to the duration of CI use. Laske et al. [43] evaluated if there is an age-related cut-off for CI treatment and concluded that CI treatment is cost-effective up to very advanced ages (~80 years). Importantly, delaying dementia (e.g., by treating hearing loss) can substantially reduce the cases of dementia, thereby reducing the costs for care and services [7,13]. The WHO has recently highlighted the enormous costs of the non-treatment of hearing loss, which can be expected to increase as the global population ages [13]. Due to its cost-effectiveness, funding for CI treatment is provided across all ages by national healthcare insurance in Germany (and in most developed economies) [44].

Several studies have demonstrated the benefits of CI treatment on people’s HRQOL and psychological well-being [19,20,21,22,23,24,25,26,27,28,29,30,31]. Less evidence has been published on the changes in speech perception and HRQOL after CI treatment, specifically in different subgroups of elderly CI recipients. Studies that assessed HRQOL and featured elderly participants often compared the elderly recipients’ results to those of younger (e.g., aged 20–40 years) recipients [25] or investigated a large age range [22,45]. Very few data are available on different age groups within the elderly population, including early- and long-term data of CI treatment in the elderly [33]. Unfortunately, Lenarz et al.’s [33] report on speech perception does not report on HRQOL and HRQOL is important because speech perception and HRQOL are poorly correlated [46,47].

The social and economic importance of hearing loss and the aging population in combination with the large gap between the number of elderly people who fulfil CI indication criteria and the number who receive CI treatment, calls for action. In particular, the study aimed at closing two remaining gaps in the current literature, namely: (1) more and prospective data should be added to the ongoing discussion and (2) clinical action points that can be used to improve CI treatment should be identified.

## 2. Materials and Methods

The study was conducted in accordance with the Declaration of Helsinki and approved by the Ethics Committee of the Hannover Medical School before study commencement (no. 1545-2012). Informed consent was obtained from all subjects involved in the study.

### 2.1. Participants

Forty-five people participated in this study, which was conducted from November 2012 to November 2014. All participants had postlingual, progredient hearing impairment and were 60–90 years old when they received the first CI. The group was divided into two subgroups according to age at implantation: Group 1 (*n* = 20) were 60–70 years of age; Group 2 (*n* = 25) were 71–90 years of age. Their demographic data are shown in Table 1.

### 2.2. Test Materials and Intervals

HRQOL and speech perception were both assessed at the same three intervals: preoperatively and at 3- and 12-months postoperatively. These intervals correspond to our clinical routine.

#### 2.2.1. Health-Related Quality of Life

HRQOL was assessed via the Nijmegen Cochlear Implant Questionnaire (NCIQ), which is a disease-specific instrument distinguishes three general domains: physical, psychological, and social functioning, which are further specified into 6 subdomains. The physical subdomain is specified in basic sound perception (1), advanced sound perception (2), and speech production (3). The psychological functioning domain consists of only one subdomain: self-esteem (4). The social domain is specified in activity (5) and social interaction (6). Items can be answered on a five-point response scale ranging from “never” to “always” or “no” to “good”. If a statement does not apply to a participant, a sixth answer can be given: “not applicable”. Total scores range from 0 (very poor) to 100 (optimal).

#### 2.2.2. Speech Perception

Speech perception in the quiet was evaluated with the Freiburg monosyllabic word test (MS) [48]. To evaluate the effect of CI use, speech perception was tested for the ipsilateral ear alone: Preoperative testing was conducted for the ear to be implanted in the best-aided condition. Postoperatively, speech perception was tested with the CI on.

### 2.3. Statistics

Demographic data were summarized by mean, range, and standard deviation (SD). Group comparisons for the NCIQ and speech perception data were analyzed using the same approach: For the two mixed ANOVAs, we defined Time at the three points of time (preoperatively, and 3 and 12 months postoperatively) as the within-subjects factor and Group (Group 1, Group 2) as the between-subjects factor. For post-hoc analyses, *t*-Tests were used and corrected for multiple comparisons according to Bonferroni. The Greenhouse-Geisser adjustment was used to correct for violations of sphericity.

## 3. Results

### 3.1. Health-Related Quality of Life

There was a significant main effect for Group (F(1, 43) = 4.78, *p* = 0.034). The younger group had significantly better NCIQ scores than the older group (62.68 versus 55.23, Figure 2). There was also a significant main effect for Time (F(1.669, 71.767) = 63.51, *p* < 0.001). Post hoc t-Tests for the whole group demonstrated the following results: NCIQ scores were significantly higher at 3 and 12 months postoperatively compared to preoperatively (t(44) = −9.044, *p* < 0.001; t(44) = −8.796, *p* < 0.001, Figure 3). No further increase was found from the 3- to the 12-month assessment (t(44) = −0.773, *p* = 0.443). No significant interaction between Time and Group was found (F(1.669, 71.767) = 2.278, *p* = 0.119).

### 3.2. Speech Perception

We found no main effect for Group in the speech perception tests (MS *p* = 0.226). There was a significant main effect for Time (MS F(2, 86) = 78.592, *p* < 0.001). Post hoc *t*-Tests showed the following: The results in the MS (speech perception in the quiet) were significantly higher at 3 and 12 months postoperatively compared to preoperatively (t(44) = −11.277, *p* < 0.001; t(44) = −10.159, *p* < 0.001), Figure 4.

There was no significant interaction between Time and Group for the speech test measure (F(2, 74) = 0.473, *p* = 0.625).

The results for the speech perception tests are presented in Figure 4.

## 4. Discussion

The present study aimed at closing two remaining gaps in the current literature, namely: (1) adding more and prospective data to the ongoing discussion and (2) identifying clinical action points that can be used to improve CI treatment. With this study design, early and long-term changes after CI activation can be investigated (i.e., most CI recipients did not yet participate in inpatient and/or intensive rehabilitation). Because both groups in the study had a sufficiently high *n*, possible differences in HRQOL and speech perception learning curves within the population of elderly adults could be meaningfully investigated via inferential statistics.

### 4.1. Health Related Quality of Life

Both groups experienced a significant increase in HRQOL after one year of CI use. This suggests that HRQOL in both groups increased in a similar way after CI treatment. The increase in HRQOL after CI treatment in elderly users is in accordance with published findings based on the NCIQ [23,24] and other outcome measures, such as a self-generated questionnaire [45]. When comparing the extent to which HRQOL increases in different age groups, various findings have been published. A recent study using the Glasgow Benefit Inventory did not find any differences between age groups [5]. In contrast, Olze et al. [23] found that elderly CI users experienced greater gains in HRQOL compared to younger adult users (19–67 years old). In the present study, Group 1 (aged 60–70 years) had generally higher NCIQ scores than Group 2 (aged 71–90 years), while the improvement in NCIQ scores did not differ between groups.

The differences in the published results could also be attributable to additional factors beyond speech perception, e.g., psychological, cognitive, or audiological factors. Future studies might consider hearing aid usage time prior to CI treatment, duration of hearing loss, CI usage time, amount and type of speech therapy sessions, and amount and type of speech therapy/rehabilitation sessions.

Although there are differences in data analyses and sample characteristics across studies, we concluded from the outcomes of the present study and published literature that CI use significantly increases HRQOL and speech perception across all age groups.

### 4.2. Speech Perception

Both groups’ speech perception improved in the quiet significantly after one year of CI use. Importantly, no difference between groups was found. This suggests a similar CI performance across all different age groups of elderly users and contradicts earlier findings [33].

### 4.3. General Discussion and Call-to-Action

The data from our present study provides additional evidence that elderly CI users with post-lingual deafness experience increases in HRQOL and speech perception as early as three months post-activation. These gains are maintained at one year of CI use.

Together with the cited literature, we conclude that CI treatment improves speech perception and HRQOL of elderly CI recipients. Benefits, however, are also evident outside the audiometric booth: CI treatment can be one key element for delaying dementia progression or even regaining cognitive capabilities [30,49]. Accordingly, Wick et al. [38]. recently concluded that CI treatment could facilitate the concept of healthy aging in the elderly. In our opinion, the following aspects are important to consider when talking about healthy aging and how it impacts a person’s life; increasing life expectancy and working lifetime, being able to live independently, and communication and social engagement. The fact that one-person households are becoming more common, especially amongst adults, increases the importance of these aspects [50].

Cost-effectiveness models have used similar parameters across ages. Research has found a correlation between hearing loss and cognitive decline (and dementia) [7,51]. This strongly suggests that the cost-effectiveness calculations for CI treatment in the elderly are conservative (i.e., result in the minimum cost-effectiveness). This is especially relevant considering that CI treatment could counteract or at least delay dementia. Further research is needed to understand the interrelationship between cognitive decline or dementia and hearing loss [52]. This suggests that if the influence of CI treatment on cognitive decline was factored in, the cost-effectiveness and effectiveness of CI provision would be even greater. The notion that hearing loss treatment merely affects hearing could be broadened extensively towards a holistic perspective, including the preservation of cognitive capabilities and (mental) health. This could mean shifting the perspective towards an “early” treatment in elderly people with the goal of preventing both auditory deprivation and the deterioration of cognitive abilities.

In our experience, it is not always clear when a person should be referred for CI evaluation. Several publications have discussed the barriers potential CI candidates face and how at least some barriers could be eliminated by updating and standardizing candidacy protocols [53,54]. It has been suggested that there might be value in referring a person even if they do not meet all criteria (yet) [55]. The authors argued that thereby, candidates and families would gain awareness of the technology and could be scheduled for future testing, if needed. Before going more into details about what could be improved and how, we would like to state some limitations of this publication. The improvement in speech perception and HRQOL demonstrated in this study are, like most studies, based on group data. Individual results vary and a small portion of adults become a limited user (~1%) or even a non-user (~2%) [56]. CI provision is not obligatory; people should have the possibility to take an informed decision.

The referral process, especially when it comes to elderly candidates, is a large barrier to CI provision in elderly candidates [54,57,58]. D’Haese et al. [13] argued that a main barrier to CI treatment is that professionals and potential candidates regard hearing loss a natural consequence of aging. This lack of awareness is a key barrier to the access of care and highlights the necessity of raising awareness for CI treatment as an option for elderly candidates [13].

In our experience, we can summarize the following points:Decision-making towards CI treatment relies on good information and support;Important aspects, particularly for the elderly, are information on electric hearing, technological aspects, anesthesia, and comorbidities;So far, referral to CI centers is still unclear and delayed; information flow towards prospective candidates is not optimal;On top of receiving information from professionals, patient associations could add a more personal perspective and serve as role-models, especially as Laplante-Lévesque and Thorén [59] concluded that information on hearing loss on the internet has only poor readability.

In our opinion, an important key measure to improve the referral process is increasing the awareness of CI treatment in general practitioners, hearing health care professionals, hearing aid acousticians, and patient associations [58]. This includes counselling people on how hearing loss impacts hearing, quality of life, and most importantly, the relationship between hearing loss and cognitive decline. Additionally, potential concerns towards CI treatment (e.g., fear of surgery) need to be considered. It may also be of benefit to include family members when counselling elderly patients. Hearing loss treatment could help manage and reduce the extensive care costs associated with dementia and other comorbidities. This complicates cost-effectiveness calculations and political discussions, especially in countries such as Germany, where care costs are covered by different bodies. Health care costs (e.g., CI treatment) are covered by the Ministry of Health, while the financial support for care (e.g., elderly home, in-home nursing staff, etc.) is covered by the Ministry of Labor and Social Affairs. This would mean that strategies evaluating the hearing loss treatment need to become more interdisciplinary from the medical perspective [7] but also from a political perspective.

With the current demographic trends, there will be more people living into their 80s and beyond. This increase in average age makes it more likely that people will live at least parts of their lives with chronic illnesses (e.g., hearing loss). This takes us back to the question of how to treat hearing loss in the elderly population appropriately and how societies can most sensibly respond to hearing loss and other chronic illness in an aging population.

## 5. Conclusions

Two groups of elderly cochlear implant (CI) recipients experienced significant increases in their hearing-related quality of life (HRQOL) and in speech perception after 3 months of CI use and maintained at 12 months of CI use. These results underscore the ability of CI use to not only improve hearing results but also to improve the lives of elderly users.

Further, we would like to use these results, and those of the numerous published papers to issue a call to action: hearing loss is common in elderly people; however, those experiencing it need not suffer it, just as elderly people with osteoarthritis can receive a prosthetic hip and walk again, elderly people with severe to profound sensorineural deafness can obtain a CI and hear again. To this end, it would be beneficial if (1) professionals counselled candidates on the holistic benefits of CI use and (2) information sources available to perspective candidates were better adapted for a lay audience. Further, as regards to funding bodies, the cost-effectiveness of CI provision in elderly candidates is very likely underestimated because it does not factor in the potential effect of CI use on preventing or impeding cognitive decline.

## Figures and Tables

**Figure 1 jpm-12-01658-f001:**
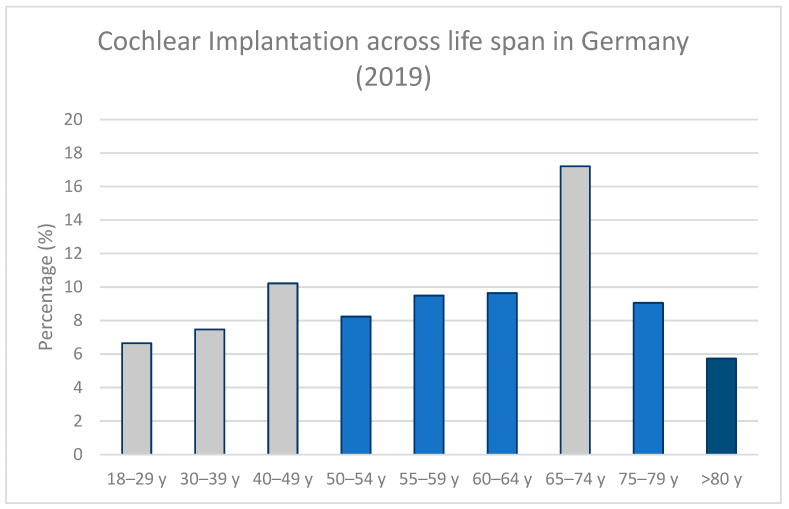
Number of new cochlear implantations per age group (18 years and older) in Germany in 2019 (according to the INEK aG-DRG Report for Germany). Approximately 84% of new CI recipients in Germany (in 2019) were adults. As can be observed, the distribution is equal across age. NB: the width of the age group ranges differs (i.e., half of the age ranges have a width of 5 years (light blue) while the other half has an age range of 10 years (grey); finally, all people above 80 years of age are grouped).

**Figure 2 jpm-12-01658-f002:**
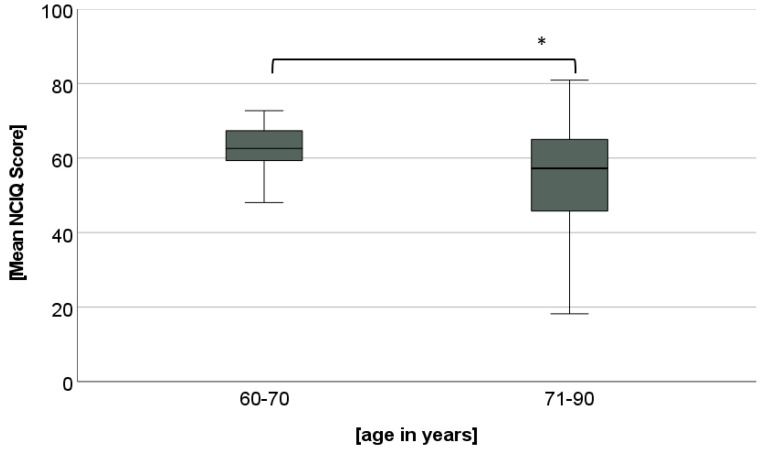
Mean scores of the Nijmegen Cochlear Implant Questionnaire (NCIQ) for each subgroup. Small circles with number indicate outliers. * means significant differences *p* ≤ 0.05.

**Figure 3 jpm-12-01658-f003:**
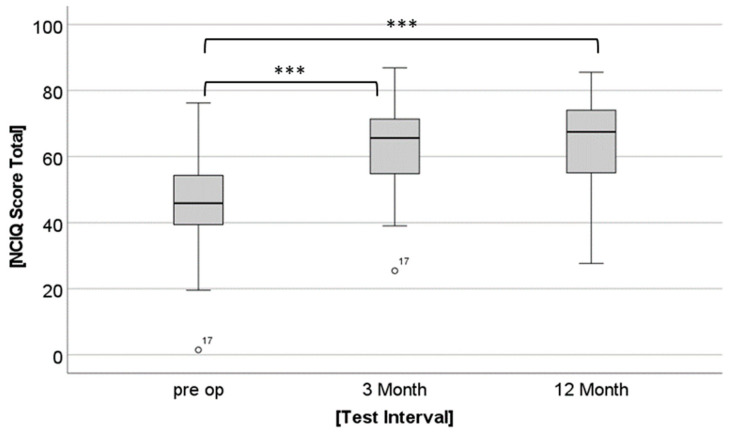
Mean scores of the Nijmegen Cochlear Implant Questionnaire (NCIQ) for the combined groups at the three test intervals (preoperative and 3 and 12 months postoperative). Small circles with number indicate outliers. ° means outliers, *** means significant differences *p* ≤ 0.001.

**Figure 4 jpm-12-01658-f004:**
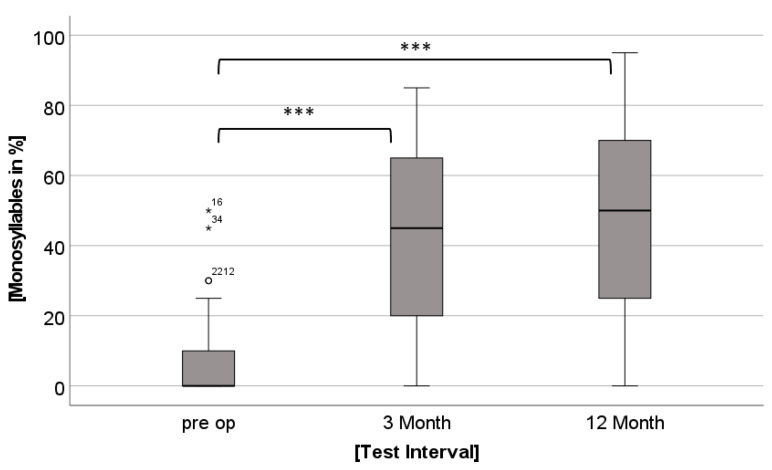
Monosyllables (MS) scores at each test interval. Small circles with numbers indicate outliers. ° means outliers, * means significant differences *p* ≤ 0.05, *** means significant differences *p* ≤ 0.001.

**Table 1 jpm-12-01658-t001:** Demographic information for each group. PTA4 = mean pure tone average at 0.5, 1, 2, 4 kHz.

	Group 1	Group 2
N	20	25
Age at implantation (in years; mean, SD, min–max)	65.82 ± 2.3; 62–70	76.87 ± 4.2, 71–86
Progredient deafness	13	20
Acute deafness	7	5
Number of additional handicaps	5. 1 each: pacemaker, limited mobility, depression, Brown-Sequard-Syndrome, Morbus Wegener	10. limited mobility: *n* = 3, pacemaker: *n* = 2, diabetes: *n* = 2, vertigo: *n* = 1, Morbus Meniere: *n* = 1, single-sided blindness: *n* = 1
Duration of HL (in years mean, SD, min–max)	28.8 ± 21.0, 1.1–64.9	27.9 ± 20.6, 1.4–77.6
Duration of deafness (in years; mean, SD, min–max)	8.5 ± 15.9, 0.2–63.5 (3 missing values)	8.6 ± 10.9, 0–38.2 (4 missing values)
Etiology		
Unknown	4	11
Sudden hearing loss	7	6
Otitis media	3	1
Genetic	3	0
Otosclerosis	0	2
Other	3	5
CI manufacturer		
Advanced Bionics	6	7
Cochlear	10	13
MED-EL	4	5
Contralateral hearing loss (PTA4):		
0–40 dB	5	0
40–60 dB	1	2
>60 dB	14	23
Contralateral ear:		
Hearing aid	9	20
Normal hearing	4	0
Untreated hearing loss	6	5
Missing data	1	/

## Data Availability

The data presented in this study are available on reasonable request from the corresponding author.

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
