# Peer review of "Do Not Go Gentle into That Deaf Night: A Holistic Perspective on Cochlear Implant Use as Part of Healthy Aging"

_jpm, 2022, doi:10.3390/jpm12101658_

Round 1
Reviewer 1 Report
This is an important study that is well-presented and very easy for the non-expert to follow. The results and discussion are significant, yet appropriately confined to the results of this particular study. Several intriguing and valid explanations for findings were recognized (such as the old age of the protocol and potential technological advancements that could explain better results than past studies.) There just appears to be a typo (is vs. it) on line 233.
Reviewer 2 Report
The manuscript describes a study intended to examine quality of life and speech perception in post-lingually-deafened patients who received cochlear implants (CI) relatively late in life. These results are presented with a minimum of detail, so that an interested reader would find it difficult or impossible to evaluate what was done. The limited information that is available suggests that the speech perception experiment in particular was superficial at best. One experiment (speech perception in noise) is mentioned several times in the text, but no results can be found. It is also stated the study provided information that might "clinical action points" related to CI use. The suggestions presented here appear to be derived from published literature. Overall, there's a long Introduction and a long Discussion, with not much in between. As a result, this report has more in common with a literature review than a data-based scientific report.
Specific comments:
line 49, "... best treatment ...": The best treatment depends on details of the hearing loss. Failing to at least mention hearing aids and other interventions here is a great oversimplification.
Figure 1 and accompanying text: Please clarify whether the numbers represent existing CI cases, new cases, or sometimes one, sometimes the other.
line 87: "Shallower", compared to what?
Section 2.2.2: The speech-perception measurements should be described in more detail.
Section 3.1: The Methods did not mention that the NCIQ was administered more than once; that should be added.
Figure 4, Table 2: Earlier text mentioned measurements of speech perception in noise, but no results are presented.
Funding, Acknowledgements: From what is written here, it seems that Med-El did provide financial support. While I appreciate that editorial assistance provided by Med-El was disclosed, I'm not sure the disclosure goes far enough. I would feel more comfortable if there were a stronger statement about the extent to which Med-El influenced the conclusions and assertions in the text; long before I read this section, I felt that this manuscript was over the top in its enthusiasm for CIs.
Reviewer 3 Report
In this manuscript, the authors report the improvements seen in the quality of life and speech perception in two elderly groups who received cochlear implants late in their lives. This is a very interesting study and will be of interest to researchers and clinicians in the field.
Specific comments:
1. Can the authors clarify to which assessment (HRQOL/Speech perception) the Greenhouse-Geisser correction was applied?
2. lines 247 & 322: Results presented (figs. 3 & 4, lines 179-80) do not suggest that HRQOL or speech perception score was " further enhanced" or "increased" from 3 months- 12 months.
General comments:
1. line 49- why is "re" in parenthesis?
2. line 233- should be "While it..."
Reviewer 4 Report
45 patients were operated with CI 8-10 years ago and followed with QOL score and speech perception for one year. No differences were found between younger 60-70 yrs and older 71-90 yrs patients. The authors discuss about the usefulness of CI in an aging population.
The data presentation of the paper could be more efficient, fig 2+3 could be combined and fig 3 + table 2 joined in one fig. The largest part of the paper is speculative on the subject of societal use of CI, based partwise on litterature. But if it is a litterture review as well, we do not know.
Round 2
Reviewer 4 Report
The authors have answered the questions posed by the referees.